# Value of Verbal Autopsy in a Fragile Setting: Reported versus Estimated Community Deaths Associated with COVID-19, Banadir, Somalia

**DOI:** 10.3390/pathogens12020328

**Published:** 2023-02-15

**Authors:** Tahlil Abdi Afrah, Lilly M. Nyagah, Asma Swaleh Ali, Mary Karanja, Hassan W. Nor, Solomon Abera, Ali Sh Mohamed, Mohamed Ahmed Yusuf Guled, Mohamed Mohamud Hassan Biday, Majdouline Obtel, Sk Md Mamunur Rahman Malik

**Affiliations:** 1College of Health Sciences, Benadir University, Mogadishu, Somalia; 2Somalia Country Office, World Health Organization (WHO), Mogadishu, Somalia; 3Department of Public Health, University Mohammed V of Rabat, Rabat, Morocco

**Keywords:** COVID-19, mortality, cause of death, verbal autopsy, Somalia

## Abstract

Background: Accurate mortality data associated with infectious diseases such as coronavirus disease 2019 (COVID-19) are often unavailable in countries with fragile health systems such as Somalia. We compared officially reported COVID-19 deaths in Somalia with COVID-19 deaths estimated using verbal autopsy. Methods: We interviewed relatives of deceased persons to collect information on symptoms, cause, and place of death. We compared these data with officially reported data and estimated the positive and negative predictive values of verbal autopsy. Results: We identified 530 deaths during March–October 2020. We classified 176 (33.2%) as probable COVID-19 deaths. Most deaths (78.5%; 416/530) occurred at home and 144 (34.6%) of these were attributed to COVID-19. The positive predictive value of verbal autopsy was lower for home deaths (22.3%; 95% CI: 15.7–30.1%) than for hospital deaths (32.3%; 95% CI: 16.7–51.4%). The negative predictive value was higher: 97.8% (95% CI: 95.0–99.3%) for home deaths and 98.4% (95% CI: 91.5–100%) for hospital deaths. Conclusions Verbal autopsy has acceptable predictive value to estimate COVID-19 deaths where disease prevalence is high and can provide data on the COVID-19 burden in countries with low testing and weak mortality surveillance where home deaths may be missed.

## 1. Introduction

Since the World Health Organization (WHO) declared coronavirus disease 2019 (COVID-19) a pandemic on 11 March 2020, this disease has affected everywhere globally and Somalia is no exception [1,2,3]. Somalia reported its first case of coronavirus disease 2019 (COVID-19) on 16 March 2021 in a student returning from abroad. Since then, and until 31 March 2022, the country had reported 26,410 cases including 1361 deaths. The number of deaths reported is low compared with neighboring countries [4]. These low figures may be because of weaknesses in the health system in Somalia including limited or lack of testing, inaccurate test results, limited access to care, and the absence of facility- or community-based mortality surveillance systems in the country [5,6]. A critical characteristic of an infectious disease such as COVID-19 is its severity, the ultimate measure of which is its ability to cause death [7].

The severity of the COVID-19 pandemic in a fragile setting such as Somalia cannot be measured by the number of deaths reported because they could be grossly underreported and undercounted, as evidenced in most African countries [8]. Cases and deaths may be under detected because of limited testing capacity or inaccessibility of health facilities [9,10]. Lower COVID-19 mortality rates in Africa have been attributed to the younger mean age of populations and lower life expectancy before COVID-19 [2]. However, it has also been suggested that the low death rates in Africa may be attributed to a paucity of data [8,11] rather than fewer actual deaths.

It has been estimated that less than half of all deaths were officially registered globally, even before the pandemic [12,13]. Therefore, countries need to use other approaches to gather data on the full effect of COVID-19 and to inform the COVID-19 response. The difference between mortality during a pandemic and the usual mortality observed for the same period in previous years, is referred to as excess mortality. Estimating excess mortality resulting from an infectious disease requires a multipronged approach and retrieval of data from different sources including health facilities, mortuaries, and communities [14,15,16,17]. Furthermore, estimating excess mortality requires complete all-cause mortality data for previous years.

The verbal autopsy is a technique used to ascertain the cause of death occurring outside the health system by interviewing caregivers of the deceased. A verbal autopsy uses a detailed questionnaire to obtain information on the symptoms, signs, and other relevant events during an illness that resulted in death and has been used to complement routine sources of mortality data [10,18,19] and, despite its limitations, has been considered for use in COVID-19 mortality reporting [10]. In countries with fragile health systems and weak or non-existent mortality surveillance systems, understanding the true extent of mortality associated with infectious diseases such as COVID-19 is challenging. As most deaths during an epidemic are likely to occur in the community because of poor health-seeking behavior, the use of verbal autopsies to predict probable causes of death is useful and can provide early indications of the impact of the disease.

In this paper, we describe the use of a modified verbal autopsy to capture deaths in the community in Somalia which could have been associated with COVID-19 and that were not picked by the country’s fragmented surveillance system. Evidence from this study will help inform policy decisions for Somalia as it considers applying alternative ways to estimate the true mortality associated with COVID-19 in the country.

## 2. Methods

### 2.1. Setting

This study was conducted in Banadir region, one of the 18 administrative regions in Somalia with an estimated four million inhabitants. The region accounted for nearly half of Somalia’s reported COVID-19 cases and deaths, according to data from our routine surveillance. Given the difficulty in obtaining accurate routine mortality data in Somalia, we developed a modified verbal autopsy for use in Banadir. We compared the number of COVID-19 deaths estimated through this verbal autopsy with the numbers reported officially by the government for the same period.

### 2.2. Data Collection

We simplified the WHO verbal autopsy tool [20] to make it easy to use and administer in the context of Somalia. Trained research assistants visited all households seeking information on deaths between March and October 2020. In households reporting one or more deaths during the period, we sought permission to interview potential respondents living with the deceased person(s) before death. We collected data using the verbal autopsy questionnaire to elicit information about possible causes of death and the presence of the main COVID-19 symptoms, specifically fever, difficulty breathing, and loss of smell and/or taste. We also collected information on the history of travel or contact with COVID-19 patients and laboratory confirmation of COVID-19, if available.

### 2.3. Official Government Reports

As is required by WHO, governments published daily reports on the status of COVID-19 in their country. These data included the number of people tested for severe acute respiratory syndrome coronavirus 2 (SARS-CoV-2; the virus causing COVID-19), test results, and number of deaths. We extracted individual-level data for all deaths in Banadir between March and December 2020 from these reports and compared them with the data obtained through the verbal autopsy.

### 2.4. Data Management and Statistical Analysis

Trained research assistants collected data on tablets using ODK (formerly Open Data Kit) forms. A deceased person who was reported to have had a cough, difficulty breathing, and loss of smell or taste before their death was categorized as a “probable COVID-19 death”. Where relatives reported that the deceased person had been confirmed as having COVID-19 before their death, we classified these cases as “reported COVID-19 death”. We combined the probable and reported deaths for the total number of COVID-19 deaths from the verbal autopsy. We used this total to compare with government reports.

At the time of the study, Somalia was already experiencing community transmission of SARS-CoV-2, hence travel history and contact were not considered in the categorization. We aggregated the data by epidemiology weeks for comparison with the other data sources and explored the sociodemographic characteristics of the deceased persons in the two sources. We used the Pearson chi-squared test to compare proportions of the sociodemographic characteristics. We also determined the sensitivity, specificity, positive predictive value, and negative predictive value of the verbal autopsy for capturing COVID-19-associated deaths by comparing with confirmed COVID-19-positive status.

### 2.5. Ethical Approval

The study was approved by the Ministry of Health of Somalia and Banadir Regional Authority. Participants gave verbal consent to provide the information for the verbal autopsy. In the analysis, data were anonymized, and no personal identifying information was included.

## 3. Results

### 3.1. COVID-19 Deaths

A total of 530 deaths were captured through the verbal autopsy between March and October 2020. Based on the WHO case definition, 176 (33.2%) of these deaths were attributable to COVID-19 because the deceased persons were reported to have had the characteristic COVID-19 symptoms (fever, loss of taste and/or smell, and difficulty in breathing). The proportion of COVID-19-associated deaths varied across the districts in Banadir, from 4.9% to 87.5% of all deaths (Figure 1).

### 3.2. Socioeconomic Characteristics

Significantly more COVID-19 deaths were reported in older people than younger people, *p* < 0.001. The highest number of deaths due to COVID-19 were recorded in April (54.7%), *p* < 0.001 (Table 1).

### 3.3. Sensitivity, Specificity and Predictive Values of Verbal Autopsy

The sensitivity of the verbal autopsy in capturing COVID-19 deaths was 86.1% (95% confidence interval (CI): 70.5–95.3%) for deaths that occurred at home and 90.9% (95% CI: 58.7–99.8%) for deaths that occurred in hospital (Table 2). Specificity of the verbal autopsy for home deaths was 67.7% (95% CI: 62.4–72.7%) and 74.7% (95% CI: 64.4–83.6%) for hospital deaths. The positive predictive value of the verbal autopsy was lower for home deaths (22.3%; 95% CI: 15.7–30.1%) than for hospital deaths (32.3%; 95% CI: 16.7–51.4%). The negative predictive value was higher: 97.8% (95% CI: 95.0–99.3%) for home deaths and 98.4% (95% CI: 91.5–100%) for hospital deaths. The sensitivity and specificity of the verbal autopsy to capture COVID-19 deaths that occurred at home remained the same for an assumed prevalence of 25% or 50% of COVID-19-associated deaths. However, the positive predictive value increased from 22.3% (95% CI: 15.7–30.1%) to 47.0% (95% CI: 42.0–52.1%) and 72.7% (95% CI: 68.5–76.5%) for an assumed prevalence of 25% and 50%, respectively (Table 2). The same was true for hospital deaths.

Verbal autopsies better predicted hospital deaths to have been caused by COVID-19 compared with home deaths: receiver operating characteristic (ROC) area 82.8% (95% CI: 72.7–92.9%) versus 76.9% (95% CI: 70.6–83.1%). Verbal autopsies would have a higher positive predictive value for home and hospital deaths at higher prevalence of COVID-19-associated deaths, but this value would remain better for hospital deaths.

### 3.4. Verbal Autopsy versus Government Reports

We compared the characteristics of individuals reported to have died using a verbal autopsy and those whose deaths were recorded by the government between March and October 2020. The deceased persons had similar sociodemographic characteristics at death in the two sources, including age and sex. Most of the deaths were among males and persons 50 years or older. The government reported 44 COVID-19 while verbal autopsies found 176 deaths during the same period.

Trends in deaths found using the two approaches followed the same pattern (Figure 2). COVID-19 deaths recorded in government reports were similar to deaths found through verbal autopsies in epidemiological week 20 but after week 22, COVID-19 deaths found through verbal autopsy were consistently higher than the government reports. The government did not report any COVID-19 deaths between week 36 and week 44.

## 4. Discussion

In this study, we describe the use and reliability of a verbal autopsy to estimate deaths in the community attributed to COVID-19 which were not necessarily recorded officially. To our knowledge, the use of verbal autopsies to ascertain the number of deaths associated with COVID-19 has not been studied in Somalia or countries with similarly fragile health systems. However, the use of verbal autopsies to capture mortality data is not new to Somalia as it has been used to ascertain maternal deaths in Bossaso [21,22].

Our study documented a discrepancy between deaths in government reports and those captured through verbal autopsies. This miscounting of the COVID-19 death toll in Somalia could be attributed to the high number of deaths occurring at home which may be missed as Somalia has no system for formal notification of deaths or mortality surveillance and a lack of regulation to report deaths [22]. A study in Morocco found that maternal mortality was underestimated because most maternal deaths occurred at home [23]. Other studies also reported gross underreporting of COVID-19 deaths across the globe [15,17,24]. Our results may provide a more accurate indication of the impact of COVID-19 in Somalia.

Low testing coverage for COVID-19 could also lead to a discrepancy in reported deaths. Government reports are based on the number of patients confirmed to have COVID-19 at the time of death. As in many countries in sub-Saharan Africa, Somalia had limited COVID-19 testing capacity with coverage ranging from <1% to 15% [9,25]. Furthermore, because of the lack of decentralized surveillance sites in Somalia the government had to rely on data from central testing laboratories.

Verbal autopsy in this study was found to have good sensitivity for both home and hospital deaths. Diagnostic accuracy is considered acceptable if the sensitivity is more than 50% for population level validation of verbal autopsies [26]. This finding differs from a study in Mozambique that found the performance of verbal autopsies to be poor [27]. However, other studies have reported the verbal autopsy to be a sensitive and valid tool for establishing deaths at the community level [14,26,27]. As reported in these studies, verbal autopsies must be contextualized given the lack of gold standards in this area. Methodological accuracy of verbal autopsy techniques should not be considered in terms of absolute validity but, more pragmatically, for their value as long as they are consistent (reproducible), comparable and adequate for the intended purpose. In our case, the purpose of using verbal autopsies was to efficiently document possible causes of death, especially in an emergency setting.

Not surprisingly, the positive predictive value was higher for hospital deaths than home deaths because confirmatory tests are done in the hospital setting which were unlikely to be done community-based settings. In addition, the difference in the positive predictive value could be attributed to the low prevalence of COVID-19 during our study as positive predictive value increased with increased prevalence. Several studies on verbal autopsies have suggested various arbitrary levels for sensitivity, specificity, and predictive values; there is no universally recognized cut-off above which the levels of these measures are considered acceptable, and the minimum levels depend on the intended use of the results [28,29,30,31]. Our results do not show that the use of a verbal autopsy is fully valid, but rather that it is a reasonable and useful approach for filling important information gaps in mortality data at the community level, thereby addressing specific public health needs that have been discussed in other studies [32,33].

There is value in using verbal autopsies during epidemics with high community-level mortality rates. With the support of WHO, Somalia has established a robust system of community health workers to respond to public health issues as they arise. These health workers were originally employed to strengthen polio surveillance, but their scope has been expanded and repurposed to include the COVID-19 response. Our study affirms the need to use pragmatic strategies such as verbal autopsy to guide programming in a country with a weak health information system.

### 4.1. Limitations

Data collected by verbal autopsy and underlying diseases were based on self-reports by a close family member of the deceased persons, and we could not verify their validity. However, the results demonstrate the usefulness of a verbal autopsy system where many community deaths would otherwise go unreported.

### 4.2. Conclusions

The verbal autopsy is an increasingly standardized technique for disease surveillance and is recognized as a feasible alternative to formal medical certification of death. WHO has called for wider use of verbal autopsies to improve understanding of the causes of mortality and the nature of mortality changes in national populations. An advantage of verbal autopsies is that deaths can be counted even without widespread testing and medical certification of cause of death. The use of key community representatives such as community health volunteers, imams, body washers, or chiefs in strengthening COVID-19 surveillance may be considered for timely and accurate data. These individuals would require training on the automated verbal autopsy tool. Other approaches such as rapid mortality surveillance systems could ensure real-time reporting of deaths at the community level.

Our study demonstrates that the verbal autopsy could be an alternative approach to assessing the causes of death in communities where many deaths occur at home. While the verbal autopsy has limitations, it provides insights on the burden of COVID-19 and associated mortality in fragile states where many deaths go unreported. The fact that verbal autopsies can be done by lay health workers as seen elsewhere supports its applicability in countries with fragile health systems such as Somalia. Our study also highlights the role of innovative approaches to complement the routine sources of mortality data in a pandemic setting and fragile states. The high number of community deaths found in our study needs further investigation.

## Figures and Tables

**Figure 1 pathogens-12-00328-f001:**
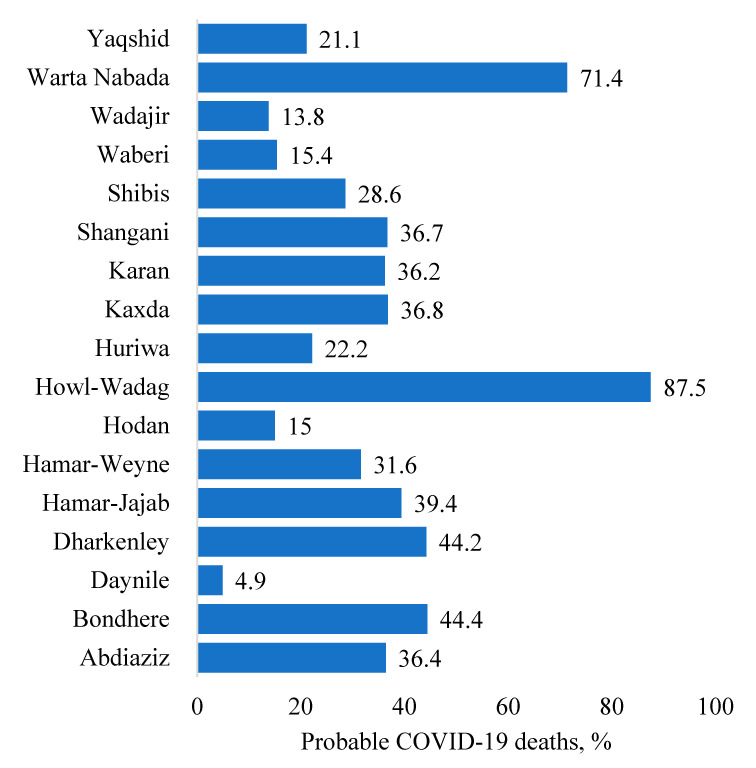
Proportion of deaths likely caused by COVID-19 based on verbal autopsy, by district, Banadir, Somalia, 2020.

**Figure 2 pathogens-12-00328-f002:**
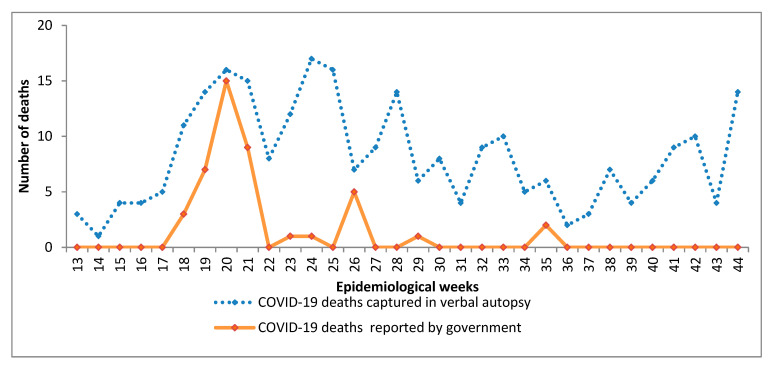
Trends in COVID-19 deaths reported in the verbal autopsy and official government reports.

**Table 1 pathogens-12-00328-t001:** Sociodemographic characteristics of the deceased persons found through the verbal autopsy, by cause of death (COVID-19 or other cause), Banadir, Somalia, 2020.

Variable	Cause of Death, No. (%)		Total, No.; *n* = 530	*p*-Value
	COVID-19; *n* = 176 (33.2) *	Other; *n* = 354 (66.8)		
Sex				0.910
Female	70 (33.5)	139 (66.5)	209	
Male	106 (33.0)	215 (67.0)	321	
Age group, in years				<0.001
0–9	0 (0.0)	24 (100.0)	24	
10–19	1 (9.1)	10 (90.9)	11	
20–29	8 (26.7)	22 (73.3)	30	
30–39	10 (20.0)	40 (80.0)	50	
40–49	16 (34.0)	31 (66.0)	47	
50+	141 (38.3)	227 (61.7)	368	
Education level				0.014
Primary or lower	115 (29.6)	273 (70.4)	388	
Secondary	37 (41.6)	52 (58.4)	89	
University	24 (45.3)	29 (54.7)	53	
Month death reported				<0.001
March	10 (32.3)	21 (67.7)	31	
April	47 (54.7)	39 (45.3)	86	
May	33 (30.3)	76 (69.7)	109	
June	29 (43.9)	37 (56.1)	66	
July	22 (30.1)	51 (69.9)	73	
August	14 (19.7)	57 (80.3)	71	
September	12 (17.9)	55 (82.1)	67	
October	9 (33.3)	18 (66.7)	27	
Place of death				0.190
Home	144 (34.6)	272 (65.4)	416	
Hospital	32 (28.1)	82 (71.9)	114	

* Includes probable and reported COVID-19 deaths but excludes deaths from non-COVID-related causes, such as accidents.

**Table 2 pathogens-12-00328-t002:** Sensitivity, specificity and predictive values of verbal autopsy to capture COVID-19 deaths occurring at home and in hospital, Banadir, Somalia, 2020.

Parameter	Estimates, % (95% CI)		
	As Reported	25% Prevalence	50% Prevalence
Death at home			
Prevalence	9.7 (6.9–13.2)	25.0	50.0
Sensitivity	86.1 (70.5–95.3)	86.1 (70.5–95.3)	86.1 (70.5–95.3)
Specificity	67.7 (62.4–72.7)	67.7 (62.4–72.7)	67.7 (62.4–72.7)
ROC area (sensitivity + specificity)/2	76.9 (70.6–83.1)	76.9 (70.6–83.1)	76.9 (70.6–83.1)
Positive predictive value	22.3 (15.7–30.1)	47.0 (42.0–52.1)	72.7 (68.5–76.5)
Negative predictive value	97.8 (95.0–99.3)	93.6 (86.6–97.1)	83.0 (68.3–91.7)
Death in hospital			
Prevalence	12.0 (6.0–20.0)	25.0	50.0
Sensitivity	90.9 (58.7–99.8)	90.9 (58.7–99.8)	90.9 (58.7–99.8)
Specificity	74.7 (64.4–83.6)	74.7 (64.4–83.6)	74.7 (64.4–83.6)
ROC area (sensitivity + specificity)/2	82.8 (72.7–92.9)	82.8 (72.7–92.9)	82.8 (72.7–92.9)
Positive predictive value	32.3 (16.7–51.4)	54.5 (44.2–64.4)	78.2 (70.4–84.5)
Negative predictive value	98.4 (91.5–100.0)	96.1 (79.1–99.4)	89.2 (55.8–98.2)

CI: confidence interval; ROC: receiver operating characteristic.

## Data Availability

Data available on request due to restrictions eg privacy or ethical The data presented in this study are available on request from the corresponding author. The data are not publicly available because its co- owned by the ministry of Health and he Banadir University.

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
