# Peer review of "Value of Verbal Autopsy in a Fragile Setting: Reported versus Estimated Community Deaths Associated with COVID-19, Banadir, Somalia"

_pathogens, 2023, doi:10.3390/pathogens12020328_

Round 1
Reviewer 1 Report
Overall, this is a well-written manuscript examining the utility of VA to better estimate deaths in the community owing to C-19. This technique documented a discrepancy of deaths as reported by government statistics. While WHO has use VA for over 50 years, there remains a paucity of information on its utility in countries such as Somalia. My comments are mostly minor. In lines 222-223, can the authors clarify their definition of “absolute validity”. Perhaps the authors might comment on how positive predictive value of VA can be improved in a future outbreaks/pandemics.
Reviewer 2 Report
With interest, I have read the ms. “The value of a verbal autopsy in fragile settings: a comparison of reported versus community estimated COVID-19-associated deaths in Banadir, Somalia” by Afrah and colleagues.
Overall, the paper offers an interesting view on the use of VA to review disease burden estimates in fragile contexts. Data included seem to be coherent with COVID-19 (symptoms, higher incidence in elderly, etc), and are therefore plausible for the study aims.
Please, find here some suggestions/comments.
Above the title, the study is categorized as study protocol, but it seems a normal research article.
It would be useful to add the simplified the WHO verbal autopsy tool as supplementary material.
Authors reported that the 28.1% of the possible COVID-19 deaths occurred in hospital setting: is it possible to add some explanation on how hospitals missed to classify these deaths as due to COVID-19 should be added?
